# Parameter Disparities Dissection for Backdoor Defense in Heterogeneous Federated Learning

**Wenke Huang[1],  Mang Ye[1,2*],  Zekun Shi[1],  Guancheng Wan[1],  He Li[1],  Bo Du[1*]**

[1] National Engineering Research Center for Multimedia Software, Institute of Artificial Intelligence,
Hubei Key Laboratory of Multimedia and Network Communication Engineering,
School of Computer Science, Wuhan University, Wuhan, China.
[2] Taikang Center for Life and Medical Sciences, Wuhan University, Wuhan, China
{wenkehuang,yemang}@whu.edu.cn
https://github.com/wenkehuang/FDCR

## Abstract

Backdoor attacks pose a serious threat to federated systems, where malicious clients optimize on the triggered distribution to mislead the global model towards a predefined target. Existing backdoor defense methods typically require either homogeneous assumption, validation datasets, or client optimization conflicts. In our work, we observe that benign heterogeneous distributions and malicious triggered distributions exhibit distinct parameter importance degrees. We introduce the Fisher Discrepancy Cluster and Rescale (FDCR) method, which utilizes Fisher Information to calculate the degree of parameter importance for local distributions. This allows us to reweight client parameter updates and identify those with large discrepancies as backdoor attackers. Furthermore, we prioritize rescaling important parameters to expedite adaptation to the target distribution, encouraging significant elements to contribute more while diminishing the influence of trivial ones. This approach enables FDCR to handle backdoor attacks in heterogeneous federated learning environments. Empirical results on various heterogeneous federated scenarios under backdoor attacks demonstrate the effectiveness of our method.

## 1 Introduction

Federated learning is an emerging collaboration learning technique [42, 119, 108, 50, 35], which allows multiple participants to perform local optimization on its own data and exchanges model parameters with a central server [67, 54, 51, 22, 28]. This federation paradigm does not require to aggregate the distributed data and obey the privacy protocol [66, 95, 73]. And the problems that come with this approach is that the central server fails to capture the client training behavior and is vulnerable to the **backdoor attacks** [23, 13, 58, 24, 56, 110]. Specifically, the evils clients makes normal predictions on benign samples and outputs the pre-defined target when the input contains a specific pattern trigger [89, 7, 17, 38, 30, 4, 98, 52]. Thus, the federated model would be implanted with the backdoor trigger pattern, which largely threatens the federated robustness. We argue that conducting the backdoor defense to erase the backdoor effect is vital for the federated reliability in the real-world application.

Driven by the serious backdoor attack, existing defense solutions could be mainly categorized into four types: Distance Difference Defense [6, 21, 93, 10, 10, 118, 27, 18], Statistics Distribution Defense [112, 25, 79, 117, 9] Proxy Evaluation Defense [48, 101, 12], and Client Side Defense [104, 123, 116, 1, 75]. The former two groups focus on detecting and mitigating malicious attack

---

*Corresponding author.

based on calculating individual distance differences or overall statistical characteristics to detect the outlier behavior. However, these two forms struggle to work under the data heterogeneous federation, where distributed data presents non-IID (independently identically distribution) and local optimization directions are dramatically distinct from each other. Therefore, they normally require the *data homogeneous assumption* for realistic settings. As regards the Proxy Evaluation Defense, they utilize the additional validation datasets with the same semantics for the ensemble distillation [87], prediction marginal contribution [102], and prediction confidence [8, 74]. Therefore, the *qualified proxy dataset* acts as a prerequisite to its feasibility and poses a huge collection obstacle in challenging scenarios, *e.g.*, medical applications [78] and financial markets [120]. Towards the Client Side Defense, it designs the client-wise regularization term to control the client updating direction such as unlearning and smoothing theory [104, 1], Hessian matrix [123], and meta-learning [75]. However, a strong assumption is that clients are willing to obey specific regularization terms and face *client optimization conflict* with existing federated optimization strategies *e.g.*, FedProx [54], MOON [51], and FPL [34]. Moreover, they *fail to resist adaptive attack*, where evils refuse to faithfully conduct the specific strategies.

Motivated by the aforementioned discussions, we are curious to rethink the Achilles heel of what malicious attack brings to federated learning systems. We assume the kernel target for malicious defense is to discriminate between benign and malicious distributions. Own to the over-parameterized characteristics of the deep neural network [26, 45], we notice that **not all parameters contribute equally to fit the target distribution**, which has confirmed soundable in the relative researches, *e.g.*, sparse and pruning strategies [49, 20, 60, 90, 88, 113]. Therefore, we argue that ***benign and malicious distributions share distinct parameter importance degree***, as confirmed in Fig. 1.

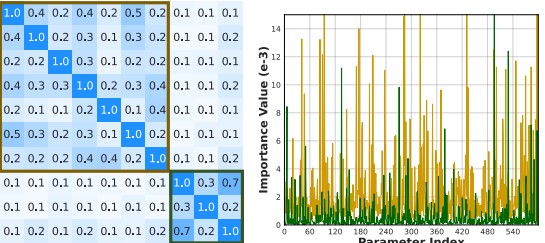

Figure 1: **Motivation**. Client parameter importance degree similarity (Left) shows difference between benign and malicious groups. The parameter important value distribution (Right) reveals that benign and malicious highlight different elements. Experiments are conducted on the Cifar-10 ($\beta = 0.5$) with three backdoor and seven benign clients.

In our work, we introduce a simple yet effective Fisher Discrepancy Cluster and Recale, abbreviated as FDCR to enhance backdoor defense ability from both client selection and parameter aggregation aspects. Preliminary, we take inspiration from the success of Fisher Information Matrix (FIM) [19, 2], which identifies the parameters information content by accessing the loss surface sharpness [76, 41, 69]. Thus, we estimate the importance of client parameters using the Fisher Information Matrix FIM on the corresponding local distribution. First, we introduce the Fisher Client Discrepancy Cluster (FCDC) to quantify the client gradient discrepancy via respective parameter importance. Clients with substantial gradient divergences are flagged as potentially malicious and excluded from the aggregation process. To be precise, After client optimization, we collect updated gradients along with their respective parameter importance, which adjusts the weight of the uploaded gradients with the expectation to accentuate crucial element updates and weaken trivial ones for the local distribution. Subsequently, we cluster the gradient updates discrepancy, identifying clients with notable discrepancies as potentially malicious. Second, we argue that during the parameter aggregation, each parameter element is allocated equal attention, which ignores the fact that parameters behave with different importance towards the target distribution. Then, we propose the Fisher Parameter Rescale Aggregation (FPRA) to rescale the element updates based on the assessed parameter importance. We argue that allocating high updating rangeability for those important parameters would increase the optimization stage and potentially weaken the trivial elements effect. For thorough examination, we conduct experiments on various heterogeneous federated scenarios [43, 46, 103], with various malicious defense solutions under the backdoor attacks [16, 84]. Experimental results reveal that ours consistently achieves stronger robustness than others. The main contributions are summarized as:

• We focus on mitigating backdoor attacks in heterogeneous federated learning. Existing solutions rely on different assumptions, *i.e.*, data homogeneity, validated samples, and faithful optimization. It motivates us to rethink the kernel behavior difference between benign and malicious clients.
• We posit that benign heterogeneous and malicious triggered distributions assign different levels of importance to parameters. To address this, we introduce the Fisher Discrepancy Cluster and Recale (FDCR), which quantifies client gradient discrepancies based on the respective parameter importance,

effectively identifying and excluding malicious participants. Additionally, we prioritize important parameter elements to enhance the optimization speed, while weakening trivial ones.

• We conduct experiments on different federated heterogeneous scenarios: Cifar-10, Fashion-MNIST, USPS, under backdoor attacks,. With ablation studies, we validate the efficacy of FDCR and the indispensability of essential modules in different setting.

## 2   Related Work

### 2.1   Federated Learning with Data Heterogeneity

Federated learning has aroused widespread interest in achieving multiple-party collaboration under security-sensitive settings [63, 50, 111]. However, its performance is limited by the distributed data, which poses non-independent and identically distribution (called data heterogeneity) [119, 53, 96, 28]. Derived from the milestone methodology, FedAvg [67], a growing body of literature has been devoted to rectifying the local drift caused by the data heterogeneity. Typical works mainly leverage the global signals such as shared model [86, 54, 51, 47, 106, 97], statistical distribution [61, 115, 122, 70, 34, 91, 92], and gradient collection [39, 22]. Some focus on self regularization [55, 114, 115, 68, 85, 32] to calibrate the biased updating direction. However, existing federated optimization methods focus on calibrating the client optimization objective to acquire a well-performing global model under the assumption of trustworthy clients. Thus, they fail to establish a defense against backdoor attacks and their effectiveness can be arbitrarily manipulated by malicious clients [29, 5, 89, 109]. In our work, we consider the client parameter importance difference and argue that malicious clients focus on fitting a largely different distribution and thus appear the different parameter important attitude. Our method is orthogonal with the above methods and is plug-and-fly to collaborate with them to improve the robustness under heterogeneous federated learning.

### 2.2   Backdoor Defense in Federated Learning

Malicious backdoor attackers bring serious threats to the federation system. To deal with backdoor attackers, existing Backdoor Defense solutions could be basically classified into four categories: **i)** *Distance Difference Defense* [6, 21, 93, 10, 10, 118, 27, 18, 33] mainly focus on distinguishing benign clients from malicious attackers via the local party updates difference and regard those significantly far from the overall direction as evils, excluded from the aggregation process. For instance, Multi Krum [6] selects the candidate gradient that is the closest to its neighboring clients. DnC [83] leverages singular value decomposition-based spectral methods for outliers detection and removal. **ii)** *Statistics Distribution Defense* [112, 25, 79, 117, 9] construct diverse statistical criteria to select and remove the evil clients. RFA [79] calculates the geometric median with an alternating minimization function. FLDetector [117] considers the historical client updates and votes for those with large discrepancies between the predicted and received client updates as attackers. Despite these advantages, the above two streams are sensitive to the degree of data heterogeneity and require complicated hyper-parameter configurations to adapt to various heterogeneous federated scenarios. **iii)** *Proxy Evaluation Defense* turn to seek help from the relative proxy datasets to conduct additional evaluation [8, 74, 31]. Specifically, FLTrust [8] collects a clean small training dataset and thus introduces Relu-clipped cosine similarity to allocate high trust scores for those reliable clients. However, the central server faces the qualified dataset collection burden, which hampers their practicability. **iv)** *Client Side Defense* [104, 123, 116, 1, 75] proposes the client-side defense based on different optimization targets, *e.g.*, clipping and smoothing operations [104]. However, these approaches necessitate that clients adhere to specific optimization regularization, rendering them vulnerable under adaptive tasks. Overall, current defense solutions demand one or more of the following: specialized hyper-parameter configuration, accessible supplementary datasets, or uniform regularization strategies. In our research, driven by the distinct characteristics of deep neural networks, we contend that parameters exhibit varied levels of importance relative to the target distribution. Therefore, we maintain that a substantial parameter importance difference between benign and malicious distributions acts as a detection signal for evils.

## 3 Methodology

### 3.1 Preliminary

Following the general federated paradigm [67, 54, 51], multiple clients collaboratively learn a shared global model $w$. For a federated system, there are $K$ clients (indexed by $k$) with the corresponding private dataset, $D_k = \{x_i, y_i\}_{i=1}^{N_k}$, where $N_k$ means the private data number for the $k^{th}$ client. At the beginning of the $t^{th}$ communication, we denote the current global model as $w^t$. Then the central server broadcasts $w^t$ to each participant as $w_k^t \leftarrow w^t$. Participating clients conduct the local optimization to fit the local distribution. Then each client uploads the optimized parameter back to the server for parameter aggregation:

$$\mathcal{L}_k(w_k^t, D_k) = \frac{1}{N_k} \sum_{\xi_i \in D_k} \mathcal{L}_{CE}(x_i, y_i), \tag{1a}$$

$$w^{t+1} = \sum_k \alpha_k w_k^t (\alpha_k = \frac{N_k}{N}). \tag{1b}$$

$N = \sum_k N_k$ denotes the overall client data scale. $\xi_i$ denotes the query sample. $\alpha_k$ denotes the pre-defined aggregation weight based on the data scale. However, some malicious clients would deliberately implant the trigger into the victim models by poisoning the training dataset [23, 13, 94, 57, 37, 99]. Specifically, we define $\mathbf{\Phi}$ as the trigger pattern and $\mathbf{m}$ as the trigger location mask. The modified backdoor instance is represented as $\widetilde{\xi} = (\widetilde{x}, \widetilde{y})$. For $\widetilde{x}$, we apply the formula $\widetilde{x} = (1 - \mathbf{m}) \odot x + \mathbf{m} \odot \mathbf{\Phi}$, incorporating the trigger pattern $\mathbf{\Phi}$ into the original instance $x$ at locations specified by the mask $\mathbf{m}$. We then alter the original label $y$ to the predefined attack target $\widetilde{y}$. Consequently, this necessitates a reformulation of the original local direction as outlined in Eq. (1a).

$$\mathcal{L}_k(w_k^t, D_k) = \frac{1}{|D_k|}[\sum_{\xi \in D_k} \mathcal{L}_{CE}(x, y) + \sum_{\widetilde{\xi} \in D_k} \underbrace{\mathcal{L}_{CE}(\widetilde{x}, \widetilde{y})}_{Backdoor}] \tag{2}$$

### 3.2 Fisher Discrepancy Cluster and Recale

#### 3.2.1 Motivation

To motivate our method, we first introduce one crucial observation of the relationship between benign and malicious clients, shown in Eqs. (1a) and (2). It reveals that benign and malicious fit the distinct distributions and naturally hold different parameter importance attitudes. Thus, in our work, we now turn to designing a strategy that can **measure the parameter importance degree for each client**. One effective way to measure the parameter importance is to consider how much changing the parameter will change the model output. We denote $p(y|x, w)$ as the output distribution over $y$ produced by a parameterized model $w \in \mathbb{R}^{|w|}$, given input $x$. One way to measure how much a change in parameters would change a model prediction is to compute KL Divergence, $KL(p(y|x, w)||p(y|x, w + \delta))$ [44], where $\delta \in \mathbb{R}^{|w|}$ is a small perturbation. As confirmed in [64, 77], we can approximate the KL divergence by its second-order Taylor series as follows:

$$\mathbb{E}_x \mathcal{L}_{KL}((p(y|x, w)||p(y|x, w + \delta)) = \frac{1}{2}\delta^T F_w \delta + \mathcal{O}(\delta^3), \tag{3}$$

where $\mathcal{O}(\delta^3)$ is short-hand to mean terms that are order 3 or higher in the entries of $\delta$. $F_w \in \mathbb{R}^{|w| \times |w|}$ is the Fisher Information Matrix (FIM) [19], which quantifies the information carried by the observable random variable about the unknown parameters $w$ on the target distribution $D$ and is formulated as the following expression:

$$F_w = \mathbb{E}_{x \sim p(x)}[\mathbb{E}_y \nabla \log \sim p(y|x, w) \cdot (\nabla \log \sim p(y|x, w))^T]. \tag{4}$$

Given this relation, it can be seen that the FIM is closely connected to how much each parameter affects the model predictions, which has been widely used in different fields [41, 86, 59, 65, 121, 81, 107]. However, due to the over-parameterized network, the computation of Fisher information is unacceptable, *i.e.*, $F_w \in \mathbb{R}^{|w| \times |w|}$. To save the computational effort, Fisher Information Matrix could be approximated as the diagonal matrix, *i.e.*, $F_w \in \mathbb{R}^{|w|}$. Furthermore, considering the expectation in Eq. (4), it is hard to draw sample $x \sim p(x)$ in most tasks, We approximate it by sampling over $N$ training samples within the dataset $D$ as follows:

$$F_w(D) \approx \mathbb{E}_{(x,y) \in D} \nabla \log p(y|x, w)^2 \in \mathbb{R}^{|w|}. \tag{5}$$

This approximation carries an intuitive explanation: A query element in $F_w$, corresponds to the average squared gradient of the output concerning a particular parameter. If a parameter significantly

influences the model output, its respective value in $F_w$ will be sizable. Consequently, we can interpret $F_w$ as a measure of the relative importance of each parameter.

### 3.2.2 Fisher Client Discrepancy Cluster

From the Eq. (5), the $F_w(D)$ quantifies the degree of importance of the parameter $w$ for the target data distribution $D$. Consequently, for each client in the federated system, we compute the parameter importance on their local data distribution $D_k$ via privately optimized model $w_k^t$. Additionally, we observe that the Fisher Information Matrix (FIM) does not remain within a fixed range, potentially leading to instability in the importance metrics across participants. To address this, we apply the min-max normalization to provide a stable description of client parameter importance as follows:

$$\mathcal{I}_k = \frac{F_{w_k^t}(D_k) - \min(F_{w_k^t}(D_k))}{\max(F_{w_k^t}(D_k)) - \min(F_{w_k^t}(D_k))} \in \mathbb{R}^{|w|}. \tag{6}$$

Then, the central server would collect the optimized model $w_k^t$ from different clients. The updated gradient for the client $k$ could be denoted as follows:

$$g_k^t = (w_k^t - w^t)/\eta \in \mathbb{R}^{|w|}, \tag{7}$$

where $\eta$ denotes the default local learning rate. Therefore, we reweight the client gradient update to highlight the parameter update with the parameter importance degree, $\mathcal{I}_k$ in Eq. (6) as:

$$\widetilde{g}_k^t = g_k^t \odot \mathcal{I}_k. \tag{8}$$

Then the aggregated global gradient update could be derived as the following form. Then, we measure the gradient update difference between the aggregated global with each client view.

$$\widetilde{g}^t = \sum_k \frac{N_k}{N} \widetilde{g}_k^t, \tag{9a}$$

$$\boldsymbol{V}_k = \sum_{v \in w} (\widetilde{g}^t - \widetilde{g}_k^t)^2 / |w|. \tag{9b}$$

Intuitively, $\boldsymbol{V}_k$ measures the gradient difference between the global and client aspects. We propose that malicious clients tailor their models to a distribution that has been artificially manipulated and consequently demonstrate a **large** discrepancy in gradient updates compared to the global aggregation. Hence, clients with pronounced disparities in their gradient updates may be indicative of malicious intent. To methodically identify such outliers, we employ unsupervised clustering to segregate the evil effect and provide a detailed comparison of popular clustering solutions in Tab. 1. We illustrate the process with five participating clients and the last two are evils:

$$\boldsymbol{V} = [\boldsymbol{V}_1, \boldsymbol{V}_2, \boldsymbol{V}_3, \boldsymbol{V}_4, \boldsymbol{V}_5]$$
$$\Downarrow \quad Cluster$$
$$= [\ \underbrace{\boldsymbol{V}_1, \boldsymbol{V}_2, \boldsymbol{V}_3}_{\text{"Benign"}}\ ,\ \underbrace{\boldsymbol{V}_4, \boldsymbol{V}_5}_{\text{"Evil"}}\ ] \quad (\frac{\boldsymbol{V}_1 + \boldsymbol{V}_2 + \boldsymbol{V}_3}{3} < \frac{\boldsymbol{V}_4 + \boldsymbol{V}_5}{2}) \tag{10}$$

We mitigate the malicious effect during the aggregation and rescale the default parameter aggregation weight $\alpha$ in Eq. (1b) as the following formulation.

$$\widehat{\alpha} = [\ \frac{\alpha_1}{\alpha_B}, \frac{\alpha_2}{\alpha_B}, \frac{\alpha_3}{\alpha_B}\ ,\ 0, 0\ ] \quad (\alpha_B = \alpha_1 + \alpha_2 + \alpha_3) \tag{11}$$

### 3.2.3 Fisher Parameter Rescale Aggregation

Furthermore, we notice that normal parameter aggregation treats all elements equally, failing to recognize their differing impacts on the target distribution. To rectify this limitation, our approach introduces the Fisher Parameter Rescale Aggregation (FPRA), designed to emphasize the parameter elements that are deemed more crucial during the aggregation phase. To be precise, we adjust the scaling of each parameter element value change, based on the importance measurement $\mathcal{I}_k$ derived from the client parameters. Additionally, to ensure that the $\mathcal{I}_k$ falls within a practical range for rescaling operation, we apply the sigmoid function to convert each parameter importance element $v \in w$ as the following formulation:

$$\widehat{g}_{k,v}^t = \underbrace{\frac{2}{1 + \exp(-\mathcal{I}_{k,v})}}_{\in[1, \frac{2e}{1+e}]} \times g_{k,v}^t \quad (\mathcal{I}_{k,v} \in [0,1]),$$

$$\widehat{w}_k^t = w^t - \eta\,\widehat{g}_k^t. \tag{12}$$

Then, based on the rescaled client parameters $\widehat{w}_k^t$ in Eq. (12) and the reallocated aggregation weight $\widehat{\alpha}$ in Eq. (11), we acquire the aggregated global parameter $w^{t+1} = \sum_k \widehat{\alpha}_k \widehat{w}_k^t$. Furthermore, we provide the detailed description in the Algorithm 1.

---

**Algorithm 1:** FDCR

---

**Input:** Communication rounds $T$, participant scale $K$, $k^{th}$ client private model $w_k^t$ and local data $D_k$
**Output:** The final global model $w^T$

**for** $t = 1, 2, ..., T$ **do**
    *Participant Side*;
    **for** $k = 1, 2, ..., K$ in parallel **do**
        $w_k^t \leftarrow$ LocalUpdating$(w^t, D_k)$ // Each client optimizes on private data
        $\mathcal{I}_k \leftarrow (w_k^t, D_k)$ via Eqs. (5) and (6) // Calculate parameter importance degree
    **end**
    *Server Side*;
    $w^{t+1} \leftarrow$ FDCR $(\{w_k^t\}_{k=1}^K, \{\mathcal{I}_k\}_{k=1}^K, w^t)$
**end**
FDCR $(\{w_k^t\}_{k=1}^K, \{\mathcal{I}_k\}_{k=1}^K, w^t)$:
**for** $k = 1, 2, ..., K$ in parallel **do**
    $g_k^t = (w_k^t - w^t)/\eta$
    $\widetilde{g}_k^t = g_k^t \odot \mathcal{I}_k$ // Reweight the client gradient updates
**end**
$\widetilde{g}^t = \sum_k \alpha_k \widetilde{g}_k^t$
$\boldsymbol{V} \leftarrow (\widetilde{g}^t, \{\widetilde{g}_k^t\}_{k=1}^K)$ through Eq. (9b) // Measure the gradient difference
$\widehat{\alpha} \leftarrow (\boldsymbol{V}, \alpha)$ by Eqs. (10) and (11) // Cluster and reallocate aggregation weight
$\widehat{w}_k^t \leftarrow (w^t, \mathcal{I}_k)$ with Eq. (12) // Rescale client parameter updates
return $w^{t+1} = \sum_k \widehat{\alpha}_k \widehat{w}_k^t$

---

## 3.3  Discussion and Limitation

**Relation wit Fisher Information Matrix Exploration**. Fisher Information Matrix (FIM) has attracted wide interest in measuring the parameter weight importance [40, 64, 36]. For example, in the continual learning field, [41, 59, 69] measures the parameter stiffness based on the historical distribution to alleviate prediction performance degradation on the previous classes. Besides, FIM is also utilized to boost the invariance representation [71, 81, 121] for domain generalization. As for federated learning, [107] argues that the initial learning phase plays a critical role in the federation, and [86] protects important parameters to enhance the federated generalization. Thus, existing works all focus on ranking the parameter importance for the target distribution. But in our work, we focus on the backdoor attack in heterogeneous federated learning and argue that backdoor attackers deliberately overfit the triggered distribution. Therefore, the backdoored model appears large parameter discrepancy with the benign client distribution. We measure the client parameter importance to reweight the client gradient updates, highlighting those clients with similar important distributions and excluding those with divergent ones.

**Clustering in FCDC Eq. (10)**. Clustering strategies have been introduced to discover natural grouping property among samples [14, 62, 3, 82, 100, 11]. For example, K-Means [62, 3] iteratively assigns points to a fixed group number. DBSCAN [15] requires to pre-define distance value. However, they are sensitive to hyper-parameter selection under different scenarios. Then, we utilize the FINCH [82], which is parameter-free and thus suitable for backdoor defense with agnostic client scale and diverse data heterogeneity. We demonstrate the superiority in Tab. 1 Specifically, we leverage the Euclidean metric to evaluate the gradient difference value $\boldsymbol{V}_k$ between any two clients and view the weight with minimum distance as its "neighbor". After clustering, we regard the set with the **larger** mean gradient discrepancy as the malicious clients and then eliminate their aggregation weights.

Table 1: **Compare Clustering** strategy in Eq. (10) for FCDC in Cifar-10 and Fashion-MNIST datasets, with $\beta \in \{0.5, 0.3\}$ and $\Upsilon = 30\%$. Please refer to the Sec. 3.3 for detailed explanations.

| FCDC | Cifar-10 | | | | | | Fashion-MNIST | | | | | |
|---|---|---|---|---|---|---|---|---|---|---|---|---|
| | $\beta = 0.5$ | | | $\beta = 0.3$ | | | $\beta = 0.5$ | | | $\beta = 0.3$ | | |
| | $\mathcal{A}$ | $\mathcal{R}$ | $\mathcal{V}$ | $\mathcal{A}$ | $\mathcal{R}$ | $\mathcal{V}$ | $\mathcal{A}$ | $\mathcal{R}$ | $\mathcal{V}$ | $\mathcal{A}$ | $\mathcal{R}$ | $\mathcal{V}$ |
| K-Means | 54.50 | 88.73 | 71.61 | 49.79 | 90.28 | 70.03 | 86.77 | 18.30 | 52.53 | 84.71 | 86.01 | 85.36 |
| DBSCAN | 65.03 | 49.46 | 57.24 | 61.91 | 38.51 | 50.20 | 87.09 | 0.70 | 43.89 | 85.16 | 0.54 | 42.85 |
| FINCH (Our) | 65.60 | 90.54 | **78.06** | 61.25 | 93.60 | **77.42** | 86.92 | 88.32 | **87.62** | 85.59 | 89.62 | **87.60** |

**Conceptual Difference**. To some extent, our approach aligns with the Distance Difference Defense paradigm. For instance, Multi Krum and FoolsGold, respectively measure squared Euclidean norm among neighboring gradients and calculate contribution similarity. Additionally, recent advancements such as DnC [83], which employs singular

Table 2: **Ablation** for $\mathcal{I}_k$ in FCDC Eq. (8) on Cifar-10 ($\Upsilon = 30\%$). Refer to Sec. 3.3.

| | $\mathcal{A}$ | $\mathcal{R}$ | $\mathcal{V}$ | $\mathcal{A}$ | $\mathcal{R}$ | $\mathcal{V}$ |
|---|---|---|---|---|---|---|
| | $\beta = 0.5$ | | | $\beta = 0.3$ | | |
| w/o $\mathcal{I}_k$ | 58.02 | 29.02 | 43.52 | 57.68 | 72.67 | 65.17 |
| w $\mathcal{I}_k$ | 63.52 | 89.56 | **76.54** | 60.20 | 93.44 | **76.82** |

value decomposition to remove outliers, and MMA [30], which utilizes multiple metrics including Manhattan, Euclidean, and Cosine distances. However, existing works regards the parameter elements as the equal importance and fail to highlight the distinct between benign and malicious clients. Therefore, we utilize the FIM to differentially highlight clients updated based on the local distribution. We conduct the experiments without considering the parameter importance degree $\mathcal{I}_k$ in Tab. 2. It appears limited performance in heterogeneous federation withou parameter importance characteristics.

**Limitation**. Our approach recognizes that benign heterogeneous and malicious triggered distributions exhibit distinct parameter importance profiles. Despite its strengths, our method primarily addresses the mitigation of the backdoor effect during the aggregation phase. Consequently, it does not effectively eliminate previously triggered parameters that persist in the model. This limitation is shared by other existing federated backdoor defense solutions that do not implement server-side optimizations, such as proxy dataset usage or post-calibration techniques (*e.g.*, finetuning, smoothing clipping [124, 104]). While our method, referred to as Fisher Parameter Rescale Aggregation, effectively identifies and prioritizes crucial parameters, the challenge of removing or unlearning already poisoned parameters remains a crucial challenge in the federation.

## 4 Experiments

### 4.1 Experimental Setup

**Datasets**. Adhere to [105, 70, 51, 30], we evaluate the efficacy and robustness on three scenarios:
• **Cifar-10** [43] contains $50k$, $10k$ images for training, validation. Each image is in $32 \times 32$ size from 10 different classes, *e.g.*, airplanes, cars, and birds.
• **MNIST** [46] is a famous digits dataset with 70,000 images in 10 classes.
• **Fashion-MNIST** [103] has 60k train and 10k test examples from 10 classes.

**Data Heterogeneity**. As for the data heterogeneity simulation, we utilize the Dirichlet distribution, $Dir(\beta)$ to simulate the label skew, as previous methods [54, 51, 115], where $\beta > 0$ is the concentration parameter to adjust the class-wise skew level. The smaller $\beta$ is, the more imbalanced the local distribution is. We set the $\beta$ as 0.5 and 0.3 for the following experimental comparison.

**Backdoor Attack**. We construct the backdoor attack based on the popular backdoor paradigm [23, 24]. The size of the trigger pattern is set to 2×6, and its location is in the top-left corner of the images. We convert the attacked label to the third class (*i.e.*, digit 2 in Digits scenario). The malicious client ratio $\Upsilon$ is 20% and 30%. The local data poisoned portion is default set as 0.5.

**Counterparts**. We compare with several Backdoor Defensesolutions in federated learning, categorized into four types. **i)** Distance Difference Defense: Multi Krum [NeurIPS'17] [6], FoolsGold [arXiv'18] [21], RLR [AAAI'21] [72], DnC [NDSS'21] [83], and MMA [ICCV'23] [30]. **ii)** Statistics Distribution Defense: Trim Median [ICML'18] [112], Bulyan [ICML'18] [25], and RFA [TSP'22] [79]. **iii)** Proxy Evaluation Defense: FLTrust [NDSS'21] [8], Sageflow [NeurIPS'21] [74], and Finetuning [80]. **iv)** Client Side Defense: CRFL [104] [ICML'21].

**Implement Details**. We provide the details from four views as follows:

Table 3: **Ablation on key components** for FDCR in Cifar-10 and Fashion-MNIST, with $\beta \in \{0.5, 0.3\}$ and $\Upsilon = 30\%$. See Sec. 4.2 for detailed discussion.

| | | Cifar-10 | | | | | | Fashion-MNIST | | | | | |
|---|---|---|---|---|---|---|---|---|---|---|---|---|---|
| FCDC | FPRA | $\beta = 0.5$ | | | $\beta = 0.3$ | | | $\beta = 0.5$ | | | $\beta = 0.3$ | | |
| | | $\mathcal{A}$ | $\mathcal{R}$ | $\mathcal{V}$ | $\mathcal{A}$ | $\mathcal{R}$ | $\mathcal{V}$ | $\mathcal{A}$ | $\mathcal{R}$ | $\mathcal{V}$ | $\mathcal{A}$ | $\mathcal{R}$ | $\mathcal{V}$ |
| | | 64.08 | 41.22 | 52.65 | 61.85 | 44.96 | 53.40 | 87.15 | 0.26 | 43.70 | 85.82 | 3.42 | 44.62 |
| ✓ | | 63.52 | 89.56 | 76.54 | 60.20 | 93.44 | 76.82 | 86.81 | 61.92 | 74.36 | 82.53 | 53.05 | 67.78 |
| | ✓ | 65.41 | 42.59 | 54.00 | 61.53 | 38.34 | 49.93 | 87.36 | 0.40 | 43.88 | 85.33 | 3.42 | 44.37 |
| ✓ | ✓ | 65.60 | 90.54 | **78.06** | 61.25 | 93.60 | **77.42** | 86.92 | 88.32 | **87.62** | 85.59 | 89.62 | **87.60** |

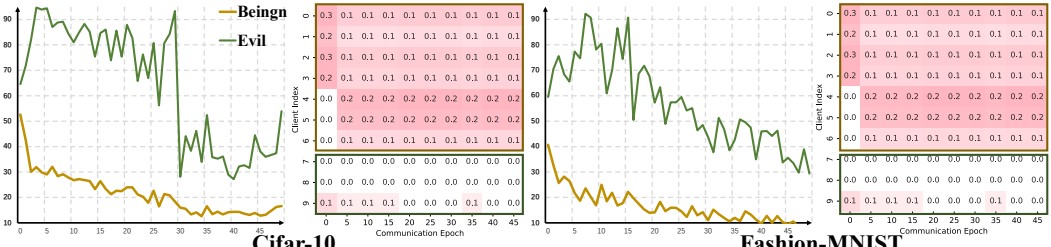

Figure 2: **Observation of gradient difference $V_k$ Eq. (9b) (Left) and aggregation weight $\widehat{\alpha}_k$ Eq. (11) (Right)** on Cifar-10 and Fashion-MNIST scenarios ($\beta = 0.5, \Upsilon = 30\%$). Backdoor attackers appear large $V_k$ and thus are gradually removed via aggregation weight $\widehat{\alpha}_k = 0$. Please see details in Sec. 4.3.

• Dataset Split: We partition the original training data into training and validation sets with a 9:1 ratio to support Proxy Evaluation Defense methods. We select a small-scale validation with the size as 256 for the methods. *e.g.*, FLTrust ad Sageflow.
• Network Structure: Following [51, 70, 34], we utilize the CNN as the backbone for Cifar-10, Fashion-MNIST, and MNIST scenarios.
• Training Setting: For a fair comparison, we follow [54, 51, 70]. We configure the communication epoch $T$ as 50, where all approaches have little or no accuracy gain with more communications. The client number $K$ is 10 for different datasets. For local training, we leverage the FedAvg [67] as the default local optimization objective. The local updating round is $E:10$ for different settings. We utilize the SGD as the local updating optimizer. The corresponding weight decay is $\eta:1e-5$ and momentum is 0.9. The local client learning rate is 0.01 in the above three scenarios. We fix the random seed to ensure reproduction and conduct experiments on the NVIDIA 3090Ti.
• Evaluation Metric: Following [67, 54, 51, 27], Top-1 accuracy is adopted for **federated benign performance**, $\mathcal{A}$ in. We further denote the **backdoor failure rate** as $\mathcal{R}$. Furthermore, we define the $\mathcal{V}$ to measure the **heterogeneity and robustness trade-off** as:

$$\mathcal{V} = \frac{1}{2}(\mathcal{A} + \mathcal{R}). \tag{13}$$

We utilize the mean performance value of the last five communication epochs as the final results.

## 4.2 Diagnostic Experiments

To thoroughly test the efficacy of crucial components of our model, we conduct a series of diagnostic studies on Cifar-10 and Fashion-MNIST datasets under the backdoor attack.

**Overall Design**. We first investigate the effectiveness of our FDCR. The results in Tab. 3 show that combining Fisher Client Discrepancy Cluster (FCDC) and Fisher Parameter Rescale Aggregation (FPRA) acquires satisfying federated benign task and backdoor removal performance that coincides with our motivation of exploiting the client parameter importance difference to mitigate the backdoor effect and enhance the relatively important parameters optimizations.

**Gradient Discrepancy and Aggregation Weight**. As shown in Fig. 2, we monitor the gradient discrepancy value $V_k$ Eq. (9b) for the arbitrary two benign and malicious clients. It shows that evils normally maintain a high $V_k$, and our method effectively detects backdoor attackers and removes the corresponding aggregation weight, *i.e.*, $\widehat{\alpha}_k = 0$.

Table 4: **Comparison with the state-of-the-art backdoor robust solutions**: in Cifar-10, Fashion-MNIST, and USPS scenarios with skew ratio $\beta \in \{0.5, 1.0\}$ and malicious proportion $\Upsilon \in \{30\%, 20\%\}$. - means the optimization failure. Best in bold and second with underline. These notes are the same as others. Please refer to Sec. 4.3 for detailed explanations.

| Methods | Cifar-10 $\beta=0.5$ | | | $\beta=0.3$ | | | Fashion-MNIST $\beta=0.5$ | | | $\beta=0.3$ | | | USPS $\beta=0.5$ | | | $\beta=0.3$ | | |
|---|---|---|---|---|---|---|---|---|---|---|---|---|---|---|---|---|---|---|
| | $\mathcal{A}$ | $\mathcal{R}$ | $\mathcal{V}$ | $\mathcal{A}$ | $\mathcal{R}$ | $\mathcal{V}$ | $\mathcal{A}$ | $\mathcal{R}$ | $\mathcal{V}$ | $\mathcal{A}$ | $\mathcal{R}$ | $\mathcal{V}$ | $\mathcal{A}$ | $\mathcal{R}$ | $\mathcal{V}$ | $\mathcal{A}$ | $\mathcal{R}$ | $\mathcal{V}$ |
| *with* malicious ratio $\Upsilon = 30\%$ | | | | | | | | | | | | | | | | | | |
| Vanilla | 64.08 | 41.22 | 52.65 | 61.85 | 44.96 | 53.40 | 87.15 | 0.26 | 43.70 | 85.82 | 3.42 | 44.62 | 95.61 | 6.13 | 50.87 | 95.75 | 2.25 | 49.00 |
| Multi Krum | 50.43 | 78.59 | 64.51 | 40.15 | 86.50 | 63.32 | 77.52 | 95.07 | 86.29 | 78.85 | 87.00 | 82.92 | 93.01 | 89.44 | 91.22 | 90.34 | 87.60 | **88.97** |
| FoolsGold | 30.71 | 23.43 | 27.07 | 53.79 | 79.53 | 66.66 | 56.10 | 0.36 | 28.23 | 69.26 | 0.81 | 35.03 | 64.70 | 11.08 | 37.89 | 39.20 | 55.41 | 47.30 |
| RLR | 64.45 | 41.19 | 52.82 | 61.89 | 44.22 | 53.05 | 87.11 | 0.41 | 43.76 | 85.75 | 3.66 | 44.70 | 95.72 | 6.56 | 51.14 | 95.68 | 2.27 | 48.97 |
| DnC | 60.01 | 90.24 | 75.12 | 56.45 | 80.07 | 68.25 | 85.40 | 91.46 | **88.43** | 83.95 | 1.99 | 42.97 | 95.26 | 77.88 | 86.57 | 94.15 | 58.79 | 76.47 |
| MMA | 54.02 | 94.26 | 74.14 | 41.69 | 61.49 | 51.59 | 79.26 | 0.10 | 39.68 | 79.31 | 0.06 | 39.68 | 93.50 | 89.80 | 91.65 | 82.62 | 1.57 | 42.09 |
| Trim Median | 46.98 | 64.53 | 55.75 | 37.45 | 50.12 | 43.78 | 10.00 | 100.0 | 55.00 | 41.59 | 71.26 | 56.42 | 44.28 | 77.22 | 60.75 | 87.79 | 2.21 | 45.00 |
| Bulyan | 41.90 | 94.19 | 68.04 | 10.00 | 100.0 | 55.00 | 10.00 | 100.0 | 55.00 | 65.02 | 98.90 | 81.96 | 83.83 | 5.31 | 44.57 | 68.87 | 1.19 | 35.03 |
| RFA | 62.92 | 32.77 | 47.84 | 61.02 | 35.51 | 48.26 | 85.66 | 0.07 | 42.86 | 85.09 | 0.43 | 42.76 | 95.68 | 4.30 | 49.99 | 95.01 | 1.97 | 48.49 |
| FLTrust | 53.46 | 79.40 | 66.43 | 41.82 | 71.27 | 56.54 | 67.45 | 5.37 | 36.41 | 70.48 | 6.90 | 38.69 | 91.50 | 66.15 | 78.82 | 92.39 | 40.70 | 66.54 |
| Sageflow | 63.71 | 35.88 | 49.79 | 61.22 | 38.63 | 49.92 | 88.05 | 1.169 | 44.60 | 86.40 | 3.52 | 44.96 | 95.78 | 6.02 | 50.90 | 95.86 | 2.81 | 49.33 |
| Finetuning | 63.12 | 44.63 | 53.87 | 62.49 | 48.96 | 55.72 | 87.42 | 4.09 | 45.75 | 85.63 | 6.11 | 45.87 | 94.65 | 9.90 | 52.27 | 94.50 | 4.31 | 49.40 |
| CRFL | 58.92 | 53.04 | 55.98 | 55.69 | 50.16 | 52.92 | 85.55 | 4.28 | 44.91 | 82.56 | 15.39 | 48.97 | 93.90 | 19.16 | 56.53 | 92.83 | 6.34 | 49.58 |
| FDCR | 65.60 | 90.54 | **78.06** | 61.25 | 93.60 | **77.42** | 86.92 | 88.32 | 87.62 | 85.59 | 89.62 | **87.60** | 95.80 | 89.34 | **92.57** | 95.44 | 77.81 | 86.62 |
| *with* malicious ratio $\Upsilon = 20\%$ | | | | | | | | | | | | | | | | | | |
| Vanilla | 65.32 | 56.11 | 60.72 | 62.23 | 48.02 | 55.12 | 87.34 | 3.44 | 45.39 | 86.25 | 12.17 | 49.21 | 95.67 | 7.46 | 51.57 | 95.93 | 30.57 | 63.25 |
| Multi Krum | 50.93 | 85.27 | 68.10 | 39.19 | 88.38 | 63.79 | 43.16 | 97.41 | 70.28 | 10.00 | 100.0 | 55.00 | 90.73 | 90.09 | 90.41 | 91.84 | 89.18 | 90.51 |
| FoolsGold | 56.24 | 95.54 | 75.89 | 50.96 | 99.47 | **75.21** | 67.13 | 17.56 | 42.34 | 66.13 | 0.163 | 33.14 | 80.76 | 94.91 | 87.84 | 47.39 | 30.12 | 38.75 |
| RLR | 64.86 | 54.86 | 59.86 | 63.51 | 50.42 | 56.97 | 87.36 | 4.34 | 45.85 | 86.06 | 8.38 | 47.22 | 95.93 | 7.77 | 51.85 | 95.65 | 29.21 | 62.43 |
| DnC | 60.87 | 84.70 | 72.78 | 57.39 | 84.78 | 71.09 | 86.49 | 89.47 | 87.98 | 84.58 | 2.94 | 43.76 | 93.75 | 49.67 | 71.71 | 95.18 | 79.91 | 87.54 |
| MMA | 52.78 | 92.62 | 72.70 | 49.60 | 82.98 | 66.29 | 78.19 | 85.76 | 81.98 | 74.62 | 16.77 | 45.70 | 93.81 | 89.64 | **91.72** | 93.21 | 89.89 | 91.55 |
| Trim Median | 46.80 | 73.69 | 60.25 | 38.23 | 90.11 | 64.17 | 81.48 | 95.68 | **88.58** | 75.10 | 86.20 | 80.65 | 92.29 | 14.22 | 53.25 | 83.31 | 30.12 | 56.72 |
| Bulyan | 42.79 | 97.87 | 70.33 | 25.37 | 76.06 | 50.72 | 77.75 | 93.60 | 85.67 | 66.58 | 94.90 | 80.74 | 87.80 | 83.87 | 85.84 | 77.82 | 48.28 | 63.05 |
| RFA | 64.69 | 46.59 | 55.64 | 61.41 | 38.83 | 50.12 | 86.61 | 0.46 | 43.53 | 86.12 | 2.68 | 44.40 | 95.51 | 16.70 | 56.10 | 94.79 | 35.40 | 65.09 |
| FLTrust | 50.54 | 81.40 | 65.97 | 43.35 | 75.66 | 59.50 | 68.06 | 16.81 | 42.44 | 71.28 | 38.53 | 54.90 | 93.43 | 89.48 | 91.46 | 92.31 | 77.26 | 84.78 |
| Sageflow | 65.29 | 44.40 | 54.84 | 61.97 | 41.24 | 51.61 | 88.29 | 8.07 | 48.18 | 86.86 | 8.65 | 47.75 | 96.31 | 9.17 | 52.74 | 95.99 | 33.93 | 64.96 |
| Finetuning | 64.02 | 58.09 | 61.06 | 63.63 | 57.57 | 60.60 | 87.28 | 13.07 | 50.17 | 85.56 | 13.74 | 49.65 | 95.06 | 15.23 | 55.14 | 94.60 | 41.85 | 68.22 |
| CRFL | 60.01 | 73.79 | 66.90 | 56.89 | 66.11 | 61.50 | 85.40 | 17.60 | 51.50 | 83.34 | 40.62 | 61.98 | 94.18 | 45.49 | 69.83 | 93.27 | 46.02 | 69.64 |
| FDCR | 65.19 | 93.59 | **79.39** | 54.49 | 93.89 | 74.19 | 85.44 | 87.47 | 86.46 | 82.67 | 84.80 | **83.73** | 91.55 | 89.79 | 90.67 | 95.63 | 90.51 | **93.07** |

Figure 3: **Comparison of federated benign performance $\mathcal{A}$ and backdoor failure rate $\mathcal{R}$ during communication** on Cifar-10 with $\Upsilon = 30\%$. FDCR appears the satisfying benign performance and backdoor failure rate. Furthermore, our method acquires stable convergence tendencies. Please see specific discussion in Sec. 4.3.

## 4.3 Comparison to State-of-the-Arts

The Tab. 4 illustrates the final metric by the end of the federated learning process with popular Backdoor Defense methods. It clearly depicts that our method achieves a satisfying performance than different counterparts on different evaluation metrics, which confirms that FDCR effectively enhances the backdoor-robust in heterogeneous federated learning. Take the result of Cifar-10 with $\beta = 0.3$ and $\Upsilon = 30\%$ as an example, our method outperforms the best counterpart with a gap of $9.17\%$ on the $\mathcal{V}$ metric. Furthermore, existing backdoor defensive methods fail to resist the backdoor attack under either the large malicious client ratio $\Upsilon = 30\%$ and serious label skew $\beta = 0.3$. It reveals that existing solutions fail to conduct the malicious discrimination selection under large-scale evils or serious data heterogeneity. We further plot both the federated benign performance $\mathcal{A}$ and backdoor

failure rate $\mathcal{R}$ during the communication process on the Cifar-10 setting in Fig. 3. We observe that FDCR presents faster and stabler convergence speed than others with different heterogeneity degrees.

## 5 Conclusion

In response to backdoor attacks in heterogeneous federated learning, we introduce the Fisher Discrepancy Cluster and Recale (FDCR), which distinguishes between benign and malicious distributions based on distinct degrees of parameter importance. We employ the Fisher Information Matrix to calculate the degree of parameter importance within client distributions and adjust the weighting of client parameter updates accordingly. Clients exhibiting large differences in gradient updates are identified as potential backdoor attackers, allowing us to mitigate their influence during the parameter aggregation process. Additionally, we prioritize and accelerate parameter elements related to the target distribution, which promotes meaningful parameter optimization and weakens the impact of non-essential elements. The effectiveness and robustness of our approach have been validated against popular counterparts in various heterogeneous federated learning scenarios. This work aims to offer a novel perspective and pave the way for future research in this field.

**Acknowledgment.** This work is supported by the National Key Research and Development Program of China 2023YFC2705700, and National Natural Science Foundation of China under Grant (62361166629, 62176188, 62225113, 623B2080). The numerical calculations in this paper had been supported by the super-computing system in the Supercomputing Center of Wuhan University.

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
