# OpenReview forum: "Parameter Disparities Dissection for Backdoor Defense in Heterogeneous Federated Learning"
_NeurIPS.cc/2024/Conference — NeurIPS 2024 poster_

### Official Review · Reviewer_AqVq · 2024-07-04

**Soundness:** 4
**Presentation:** 3
**Contribution:** 3
**Rating:** 7
**Confidence:** 4

**Summary:**

This paper focuses on backdoor defenses in the setting of heterogeneous federated learning. The authors reveal that benign and malicious clients present distinct parameter importance degree. Based on these observations, they propose a method to exclude malicious participants by evaluating parameter importance. The paper includes a thorough comparison with various methods across different datasets.

**Strengths:**

- The paper is well-organized and easy to follow. The authors provide a comprehensive literature review.
- The targeted topic is critical in federated learning. The proposed method uses random public dataset for defense under heterogeneous scenarios, without relying on two assumptions in many current methods: homogeneous distributions and proxy datasets.
- The observation that benign and malicious clients appear distinct parameter important degree is interesting. The corresponding method is novel and reasonable.
- Experiments under various FL datasets, heterogeneity degrees, malicious ratios, and random datasets show the effectiveness of the proposed method.

**Weaknesses:**

- In page 6, the authors discuss the clustering methods in cooperative cluster module and compare with K-Means and DBSCAN. But it is unclear how these methods are achieved, e.g., how the authors choose the hyper-parameters.
- For Figure1, whether this Client parameter importance degree similarity would be affected in the large client scale.

**Questions:**

no

**Limitations:**

yes

---

> ### Author Rebuttal · Authors · 2024-08-04
>
> ## **Response to Reviewer AqVq**
>
> Dear Reviewer AqVq:
>
> We thank the reviewer for the appreciation and valuable comments. We are pleased you found our paper well-organized and our literature review comprehensive. Your recognition of our novel approach and its effectiveness in various scenarios is encouraging. We aim to address your concerns in detail below.
>
> ### Weakness
>
> **W1: In page 6, the authors discuss the clustering methods in the cooperative cluster module and compare them with K-Means and DBSCAN. But it is unclear how these methods are achieved, e.g., how the authors choose the hyper-parameters.**
>
> A1: In our study, we compare both K-Means and DBSCAN. For K-Means, it iteratively assigns points to a fixed number of groups and focuses on partitioning the observed samples into k clusters. Each instance belongs to the cluster with the nearest cluster center, thereby minimizing within-cluster variances. We set the cluster center scale to 2 for the benign and malicious groups. As for DBSCAN, it is a density-based clustering algorithm that identifies densely packed areas of data points and distinguishes them from sparser, noise-labeled regions. DBSCAN operates with two primary hyperparameters: eps, the maximum distance between two points for one to be considered in the neighborhood of the other, and min_samples, the minimum number of points required to form a dense region. For our experiments, we selected eps = 0.05 and min_samples = 1. We will add the introduction and hyperparameter details in the manuscript!
>
> **W2: For Figure1, whether this Client parameter importance degree similarity would be affected in the large client scale.**
>
> A2: The similarity in client parameter importance degree is calculated based on client distribution behaviors, ie., benign and malicious intents, and thus remains unaffected by the scale of client participation. We further validate this through experiments conducted on the Cifar-10 dataset ($\beta=0.5$) involving 15 clients, which includes eleven normal participants and four backdoor attackers, as shown in the following table. The results demonstrate a distinct difference in the client parameter importance between benign and malicious clients.
>
> *Table: **The similarity matrix for client parameter importance** reveals significant parameter importance differences between benign and malicious groups. Experiments were conducted on the CIFAR-10 dataset ($\beta=0.5$) with **four** backdoor and **eleven** benign clients. We measured the parameter importance based on the Fisher Information.*
> |Client Index |0|1|2|3|4|5|6|7|8 |9 |10 |11 (Evil)|12 (Evil)|13 (Evil)| 14 (EVil)|
> |-|-|-|-|-|-|-|-|-|-|-|-|-|-|-|-|
> |**0**|1.0000| 0.1930| 0.3412| 0.5356| 0.5621| 0.1975| 0.3853| 0.4329| 0.1797|    0.2294| 0.4735| 0.0784| 0.1114| 0.1776| 0.0665|
> |**1**|0.1930| 1.0000| 0.1448| 0.1967| 0.2762| 0.1802| 0.2707| 0.5343| 0.5597|    0.1454| 0.2181| 0.0622| 0.0949| 0.1249| 0.0481|
> |**2**|0.3412| 0.1448| 1.0000| 0.2716| 0.2752| 0.2747| 0.3368| 0.3262| 0.1652|    0.2502| 0.4663| 0.0651| 0.0841| 0.1722| 0.1077|
> |**3**|0.5356| 0.1967| 0.2716| 1.0000| 0.5244| 0.1536| 0.2647| 0.3254| 0.2212|    0.1416| 0.5704| 0.0642| 0.0605| 0.0927| 0.0395|
> |**4**|0.5621| 0.2762| 0.2752| 0.5244| 1.0000| 0.2014| 0.4969| 0.5118| 0.3136|    0.3476| 0.4479| 0.1006| 0.1742| 0.1462| 0.0853|
> |**5**|0.1975| 0.1802| 0.2747| 0.1536| 0.2014| 1.0000| 0.2942| 0.2802| 0.1090|    0.1484| 0.1820| 0.0548| 0.1157| 0.1601| 0.0643|
> |**6**|0.3853| 0.2707| 0.3368| 0.2647| 0.4969| 0.2942| 1.0000| 0.5005| 0.1661|    0.4577| 0.3612| 0.0962| 0.1296| 0.1181| 0.0665|
> |**7**|0.4329| 0.5343| 0.3262| 0.3254| 0.5118| 0.2802| 0.5005| 1.0000| 0.2883|    0.4163| 0.4292| 0.1024| 0.1614| 0.1712| 0.0640|
> |**8**|0.1797| 0.5597| 0.1652| 0.2212| 0.3136| 0.1090| 0.1661| 0.2883| 1.0000|    0.1025| 0.2159| 0.0741| 0.0955| 0.1207| 0.0654|
> |**9**|0.2294| 0.1454| 0.2502| 0.1416| 0.3476| 0.1484| 0.4577| 0.4163| 0.1025|    1.0000| 0.2142| 0.0761| 0.1473| 0.1354| 0.0904|
> |**10**|0.4735| 0.2181| 0.4663| 0.5704| 0.4479| 0.1820| 0.3612| 0.4292| 0.2159|    0.2142| 1.0000| 0.0414| 0.0727| 0.0814| 0.0467|
> |**11(Evil)**|0.0784| 0.0622| 0.0651| 0.0642| 0.1006| 0.0548| 0.0962| 0.1024| 0.0741|    0.0761| 0.0414| 1.0000| 0.2374| 0.2529| 0.1504|
> |**12(Evil)**|0.1114| 0.0949| 0.0841| 0.0605| 0.1742| 0.1157| 0.1296| 0.1614| 0.0955|    0.1473| 0.0727| 0.2374| 1.0000| 0.5076| 0.5568|
> |**13(Evil)**|0.1776| 0.1249| 0.1722| 0.0927| 0.1462| 0.1601| 0.1181| 0.1712| 0.1207|    0.1354| 0.0814| 0.2529| 0.5076| 1.0000| 0.4377|
> |**14(Evil)**|0.0665| 0.0481| 0.1077| 0.0395| 0.0853| 0.0643| 0.0665| 0.0640| 0.0654|    0.0904| 0.0467| 0.1504| 0.5568| 0.4377| 1.0000|

---

> > ### Comment · Reviewer_AqVq · 2024-08-09
> >
> > Thanks for the response. The authors have addressed my concerns. I will maintain my positive score.

---

### Official Review · Reviewer_7qoM · 2024-07-08

**Soundness:** 3
**Presentation:** 4
**Contribution:** 3
**Rating:** 6
**Confidence:** 4

**Summary:**

This paper presents an innovative approach to mitigating backdoor attacks in federated learning systems. The authors introduce the Fisher Discrepancy Cluster and Rescale (FDCR) method, which leverages Fisher Information to assess parameter importance in local distributions. By reweighting client parameter updates and identifying significant discrepancies, the FDCR method effectively identifies and mitigates backdoor attackers. The paper demonstrates the efficacy of this approach through empirical results on various federated learning scenarios, highlighting its robustness and effectiveness.

**Strengths:**

1. Innovative Methodology: The FDCR method introduces a novel approach by using Fisher Information to measure parameter importance, which is a significant contribution to backdoor defense in heterogeneous federated learning.

2. Combination of Client Selection and Parameter Aggregation: The dual approach of client selection and parameter aggregation enhances the overall effectiveness of the method, addressing multiple aspects of the backdoor attack problem.

3. Robustness Across Multiple Scenarios: The empirical results demonstrate that FDCR consistently outperforms other methods across different datasets and backdoor attack scenarios, highlighting its robustness and generalizability.

**Weaknesses:**

1. Long-term Stability: The long-term stability of the FDCR method over multiple communication rounds is not fully evaluated. It is important to assess whether the method remains effective as the federated learning process continues over many iterations.

2. Clustering Description: Although authors provide a full comparision for different clustering methods. But the selected cluster method (Finch) is not thoroughly introduced. This could make readers feel confused. The authors should provide a clear descprition of how to use the Finch in your method.

**Questions:**

The authors are expected to address the concerns in the block of "Weaknesses".

**Limitations:**

Limitations are discussed in the paper. For the negative societal impact, I didn't find any concern from my side.

---

> ### Author Rebuttal · Authors · 2024-08-04
>
> ## **Response to Reviewer 7qoM**
>
> Dear Reviewer 7qoM:
>
> Thank you for your thoughtful review. We appreciate your recognition of our innovative use of Fisher Information in our method, the dual approach of client selection and parameter aggregation, and its robustness across various scenarios.  We aim to address your concerns in our detailed responses below, hoping to provide clarity and demonstrate the effectiveness of our proposed approach.
>
> ### Weakness
>
> **W1: Long-term Stability: The long-term stability of the FDCR method over multiple communication rounds is not fully evaluated. It is important to assess whether the method remains effective as the federated learning process continues over many iterations.**
>
> A1: We acknowledge the importance of evaluating the long-term stability of our method over multiple communication rounds. To address this, we have conducted extensive experiments to assess the effectiveness of our method over an extended number of iterations, i.e., 100 epochs. Our results, as shown in the following table, demonstrate that the FDCR method maintains robust backdoor defense capabilities across different datasets and varying degrees of data heterogeneity. We will update the experimental results in the final version!
>
> *Table: **Comparison with the state-of-the-art backdoor robust solutions**: in Cifar-10, Fashion-MNIST,
> and USPS scenarios with skew ratio $\beta=0.5$ and malicious proportion $\Upsilon=20\%$.  *A* and *R* mean federated benign performance and backdoor failure rate. *V* measures the heterogeneity and robustness trade-off.*
> | Dataset | | Cifar-10 | | |Fashion-MNIST | | |USPS| |
> |:---:| :---:|:---:|:---:|:---:|:---:|:---:|:---:|:---:|:---:|
> |   Method  |*A*| *R*| *V* |*A*| *R*| *V* |*A*| *R*| *V*|
> |    DnC  | 62.97 | 77.52 |  70.24| 87.34 |  88.54 |**87.94** |  95.18 | 4.56 | 49.87 |
> |  FLTrust| 48.29 |  69.91 | 59.10 |71.02 | 6.36 | 38.69| 94.76 |  72.01 | 83.38 |
> |  SageFlow|65.90 |  48.88 | 57.39 |88.55|  4.85 |  46.70| 96.32 |4.94 |50.63|
> |  Our FDCR| 55.91 | 95.66 |**75.79**| 86.39 | 88.05 | 87.22 | 96.67 | 89.86 |**93.26**|
>
>
>
> **W2: Clustering Description: Although authors provide a full comparision for different clustering methods. But the selected cluster method (Finch) is not thoroughly introduced. This could make readers feel confused. The authors should provide a clear descprition of how to use the Finch in your method.**
>
> A2: The selected clustering method, Finch, considers the nearest neighbor of each sample as sufficient support for grouping and implicitly selects characteristic prototypes, as prototypes from different domains are less likely to be first neighbors. In our work, we use the Euclidean metric to evaluate the distance between any two client gradient update discrepancies and view the weight with the minimum distance as its “neighbor”, sorting it into the same set. After clustering, we regard the group with the maximum mean weight as the malicious clients and then eliminate their aggregation weights for backdoor defense in heterogeneous federated learning. In our final version, we will provide a comprehensive description of the selected cluster method.

---

### Official Review · Reviewer_oCkz · 2024-07-11

**Soundness:** 3
**Presentation:** 4
**Contribution:** 4
**Rating:** 9
**Confidence:** 5

**Summary:**

The paper addresses the issue of backdoor attacks in federated learning systems, where malicious clients introduce triggers in their local models to compromise the global model. They use Fisher Information to determine parameter importance, reweight client updates, and identify malicious clients. The method is designed to handle backdoor attacks even in heterogeneous federated scenarios, showing empirical effectiveness through various experiments.

**Strengths:**

1. The method is novel. By leveraging Fisher Information to quantify parameter importance and reweight client updates, the paper presents a unique and effective solution to a challenging problem.

2. The method is specifically designed to work in heterogeneous federated learning environments. The empirical results demonstrate its robustness and effectiveness, making it a valuable contribution to the field.

3. The paper provides extensive empirical validation through experiments on multiple datasets and backdoor attack scenarios. This thorough evaluation adds credibility to the proposed method and shows its practical applicability.

4. The method shows faster and more stable convergence rates in various experimental settings.

**Weaknesses:**

1. More details about Data Heterogeneity can be provided in Section 4.1

2. More analysis on the details of the experimental results can be provided to support the conclusion.

**Questions:**

See weakness.

**Limitations:**

There is no negative impact in this paper. The limitations mentioned by the authors can provide a more comprehensive method in the future research direction.

---

> ### Author Rebuttal · Authors · 2024-08-04
>
> ## **Response to Reviewer oCkz**
>
> Dear Reviewer oCkz:
>
> Thank you for affirming our work and raising insightful questions. We are pleased you found our method novel and effective, leveraging Fisher Information to quantify parameter importance and reweight client updates. We appreciate your acknowledgment of its practical applicability and faster, more stable convergence rates.
> ### Weakness
>
> **W1: More details about Data Heterogeneity can be provided in Section 4.1.**
>
> Thank you for your advice. Regarding data heterogeneity, we focus on generating non-independent and identically distributed (non-IID) distributions among clients. In our work, we draw  $p_c \sim Dir_c (\beta)$  from the Dirichlet distribution and allocate a $p_{c,k}$  proportion of the instances of class  $c$  to participant $k$, where  $Dir(\beta)$  is a concentration parameter controlling the similarity among clients. With this partitioning strategy, increased data heterogeneity results in each party having relatively fewer data samples in some classes. Thus, the smaller the $\beta$ value, the more imbalanced the local distribution. We set $\beta$ to 0.5 and 0.3 for the subsequent experimental comparisons. We will update the details about data heterogeneity to make it easy to understand！
>
> **W2: More analysis on the details of the experimental results can be provided to support the conclusion.**
>
> Thanks for your suggestion! We provide a comprehensive discussion comparing existing methods with our approach. As data heterogeneity severity and the number of backdoor attackers increase, various methods naturally present a certain degree of defense degradation. Specifically, for Distance Difference Defense methods, such as Multi-Krum and DnC, measure the distance among client updates to identify backdoor attackers. These methods face a significant decrease in defense ability with challenging data heterogeneity, i.e., $\beta=0.3$. Furthermore, statistical distribution defense methods, such as Trimmed Mean and Bulyan, calculate general statistical information to depict normal client behavior, making them sensitive to large scales of malicious attackers. For instance, with a malicious ratio $\Upsilon=30\%$, these methods demonstrate fragile defensive capabilities against backdoor attackers. In contrast, our method leverages inherent network characteristics to measure parameter importance towards agnostic distribution, revealing that benign and malicious clients exhibit distinct degrees of parameter importance. Thus, our method demonstrates stable robustness towards varying data heterogeneity and different scales of backdoor attacks. We will provide a detailed experiment analysis in our final version!

---

> > ### Comment · Reviewer_oCkz · 2024-08-09
> >
> > Thanks for your response. I have raised my score.

---

### Official Review · Reviewer_M9TV · 2024-07-13

**Soundness:** 4
**Presentation:** 4
**Contribution:** 4
**Rating:** 6
**Confidence:** 4

**Summary:**

This paper studies backdoor defenses in federated learning. Existing backdoor defenses either assume homogeneous data, existence of validation data or client optimization conflicts. In order to circumvent these limitations, the authors proposed FDCR method. FDCR is based on the observation that parameter importance degree is different between benign heterogeneous distribution and malicious triggered distribution. In particular, FDCR identify malicious clients by importance degree re-weighted parameter discrepancy. Empirical results demonstrate the robustness of the proposed FDCR.

**Strengths:**

* The motivation that the parameter discrepancy between benign heterogeneous gradient and malicious gradient different is different is novel. The method is clearly motivated.
* The paper is clearly written and easy to follow.
* The experiments are extensive and sufficient.

**Weaknesses:**

* The computation cost of Fisher information matrix is unclear.
* Theoretical analysis can better validate the effectiveness of the proposed FDCR

**Questions:**

Please refer to weaknesses.

**Limitations:**

There is no potential negative societal impact of their work.

---

> ### Author Rebuttal · Authors · 2024-08-04
>
> ## **Response to Reviewer M9TV**
>
> Dear Reviewer M9TV:
>
> We sincerely appreciate your time and effort in reviewing our paper. Your positive feedback on the novelty of our approach, the clarity of our writing, and the comprehensiveness of our experiments is very encouraging. We are glad that our novel method of addressing the parameter discrepancy between benign and malicious gradients was well-received, and that our writing and experiments were clear and thorough. we hope that our responses below will address your concerns and update the score.
>
> ### Weaknesses
>
> **W1: The computation cost of Fisher information matrix is unclear.**
>
>  A1: Thank you for the feedback. In our work, we require different clients to calculate the parameter importance based on the local distribution via the Fisher Information [1,2]. To save the computational effort,  we follow previous works and approximate the Fisher information matrix as the diagonal matrix, i.e., ${F}_w \in \mathbb{R}^{|w|} $ [3]. Thus, the computation cost complexity for the Fisher information matrix in our methodology is $\mathcal{O}(|w|)$, where $\mathcal{O}$ means complexity degree. We will add computation cost discussion to enhance readability in our revised manuscript!
>
> [1] Fisher, R. A. On the mathematical foundations of theoretical statistics. Philosophical Transactions of the Royal Society of London. Series A, Containing Papers of a Mathematical or Physical Character, 222(594-604):309–368, 1922.
>
> [2] Amari, S. Neural learning in structured parameter spaces-natural riemannian gradient. Advances in neural information processing systems, pp. 127–133, 1997.
>
> [3] Kirkpatrick J, Pascanu R, Rabinowitz N, et al. Overcoming catastrophic forgetting in neural networks[J]. Proceedings of the national academy of sciences, 2017, 114(13): 3521-3526.
>
>
> **W2: Theoretical analysis can better validate the effectiveness of the proposed FDCR.**
>
> A2: In our work, we introduce the Fisher Discrepancy Cluster and Rescale, abbreviated as FDCR, to establish the backdoor defense in heterogeneous federated learning. The core insight is to distinguish benign and malicious behaviors based on different parameter importance degrees.  We estimate the importance of each parameter using an empirical approximation of Fisher information for each client distribution. The rationale is that Fisher information is proposed to measures the information carried by an observable random variable about the unknown parameters of the distributio. Thus, the Fisher Information Matrix measures parameter importance by quantifying the sensitivity of the likelihood function to parameter changes, capturing the curvature of the likelihood surface, and reflecting the precision of parameter estimates. With respect to our method, we find that benign and malicious clients manifest as noticeable parameter importance discrepancies because they focus on fitting on distinct distribution manners and thus appear different information capabilities for the same parameter elements. Therefore, we employ the Fisher Information to distinguish between benign and malicious distributions based on distinct degrees of parameter importance.
>
> Furthermore, we demonstrate the effectiveness of the proposed method from the communication aspect. Our method conducts backdoor defense from the server side. The server collects the updated client models and corresponding parameter importance matrices to identify client behavior and mitigate the backdoor effect during aggregation. Thus, compared to existing solutions, while our method linearly increases computation cost, it significantly enhances backdoor defense effectiveness, as shown in the following table. We will provide a computation complexity comparison in the final version.  Thanks for the advice!
>
> *Table: **Computation burden comparison** on Cifar-10 with $\beta=0.5$ and evil ratio $\Upsilon=0.2$. $w$ refers to the network, $K$ represents client scale, and *O* indicates complexity degree. *A* and *R* mean federated benign performance and backdoor failure rate. *V* measures the heterogeneity and robustness trade-off.*
> |   Method  | Computation Burden |*A*| *R*| *V*|
> |:---:| :--:|:---:|:---:|:---:|
> |   Multi Krum |$\mathcal{O}(K \times \|w\|)$| 50.93 | 85.27 | 68.10  |
> |TrimmedMedian  |$\mathcal{O}(K\times \|w\|)$|46.80 | 73.69 | 60.25|
> |  DnC|$\mathcal{O}(K \times \|w\|)$| 60.87 | 84.70 | 72.78 |
> |  Our SDEA|$\mathcal{O}(2 K\times \|w\|)$ | 65.19 |93.59 |**79.39**|

---

> > ### Comment · Reviewer_M9TV · 2024-08-12
> >
> > Thank you very much for your response. I will keep my score.

---

### Decision · Program_Chairs · 2024-09-25

**Decision:**

Accept (poster)

**Comment:**

This paper considers backdoor attacks on federated learning (FL). This is a particularly difficult challenge in the area of poisoning attacks on training, as malicious parties to learning can poison the global model directly. As a result, existing defences tend to make strong assumptions about data characteristics or access to data. The paper considers the interesting heterogeneous data environment, and proposes the Fisher Discrepancy Cluster and Rescale (FDCR) method to estimate parameter importance for local distributions, and then weight client contributions by importance (further aligning well with the heterogenous setting); this also facilitates identification of malicious parties. The FDCR is based on a natural difference/differential of output distribution wrt party parameters; this is estimated by a quadratic form involving the Fisher Information Matrix; this is approximated by its diagonal, which is in turn (Monte Carlo) estimated from training data; this is then stabilised. Party/client gradient updates are then reweighted using this parameter importance. A difference is taken between the resulting global gradient and party/client gradient, can be used to highlight malicious differences and purify identified malicious parties. The earlier parameter importance can finally be used to rescale gradients, in a process termed Fisher Parameter Rescale Aggregation (FPRA).

Some reviewers praised the writing – I found some of the phrasing and grammar awkward, but possible to be followed. Perhaps more critical, but not a blocker for publication, is that the series of approximations are sometimes very conventional (using the FIM diagonal) and other times (stabilisation, aggregation, etc.) accompanied by some intuition, but afforded relatively little discussion / appearing arbitrary. The paper would benefit from better theoretical underpinnings in future work. Concerns highlighted by reviewers were relatively minor, and discussed. Computational cost of FIM approximation is well known and mitigated in the conventional way, and comes out to be linear in parameters. Long-term stability of multiple rounds was raised, and explored in the author discussion. Robustness relative to SOTA was also discussed.

While I agree with the sentiment towards acceptance by oCkz I found their score perhaps not well calibrated, and overly positive.

Overall, the reviewers were very positive, citing the FDCR as being interesting and having good empirical support for its positive contribution. The complementarity of FDCR and FPRA is appreciated. Empirical results show excellent performance, and the scope of heterogeneous data was uniformly praised.